# CL-Gen: An Inference-Time Iterative Optimization Framework for Reference-Consistent Image Generation Based on Closed-Loop Control

## Abstract

Controllable image generation technology enables precise content synthesis based on user-provided reference conditions, garnering significant research attention. However, existing methods still face significant challenges in maintaining reference consistency, as they lack the observation and error correction for the reference consistency of generated images. Inspired by the concept of closed-loop systems in control theory, we propose a framework that enhances reference consistency through an iterative optimization scheme during inference time. It takes the image generation model as the control plant, observes and feeds back the actual generation state, and then adjusts the input of the generation model through a modified PID (Proportianl Integral Derivative) controller to enhance reference consistency. This framework supports a variety of controllable generation methods and different types of control conditions. Moreover, it is easy to implement, requiring only the addition of a few lines of code without the need for extra training. We validate the application of this framework in three key tasks: identity-preserving portrait generation, pose-controlled generation, and depth-controlled generation. For identity-preserving portrait generation, our method improves facial similarity by 12.07%. For pose-controlled and depth-controlled generation, errors are reduced by 32.64% and 33.49%, respectively. This work not only provides a solution for reference-consistent image generation but also offers a new perspective: controllable image generation can be conceptualized as a control problem, wherein control theory is amenable to application for performance optimization. Our code will be released upon publication.

## 1 Introduction

Image generation technology is driving digital transformation by boosting production efficiency, lowering innovation barriers, and unlocking new business opportunities. Text-to-image diffusion models (Ramesh et al., 2022; Rombach et al., 2022; Black Forest Labs, 2024) have advanced the field, enabling high-quality image generation from text prompts. Recent progress has added controllable generation via visual references, addressing text prompt limitations. These breakthroughs have improved traditional applications like style transfer (Zhang et al., 2023b; Wang et al., 2023), image enhancement (Wang et al., 2025; Jiang et al., 2023), and super-resolution (Li et al., 2022; Gao et al., 2023), while fostering innovative uses such as aesthetic QR codes (Wu et al., 2024; Cui et al., 2024) and virtual try-on (Dam et al., 2024; Choi et al., 2024; Yang et al., 2024b).

Many applications require strict consistency between the generated outputs and the reference conditions. A prominent example is ID-preserving portrait generation (Ye et al., 2023; Wang et al., 2024; Li et al., 2024d) , where generated images need to maintain subject identity through precise replication of facial features and other biometric characteristics. However, existing methods often fail to guarantee reference consistency, deviating from user expectations. We draw direct inspiration from control theory, where closed-loop systems leverage continuous feedback to force a system's output to converge stably to a target value. This paradigm leads to pivotal questions: **can controllable image generation be formulated as a control problem and can control theory be applied to optimize it?** Specifically, is it feasible to construct a closed-loop system to make the features of the generated images converge toward the conditional input, thereby improving reference consistency?

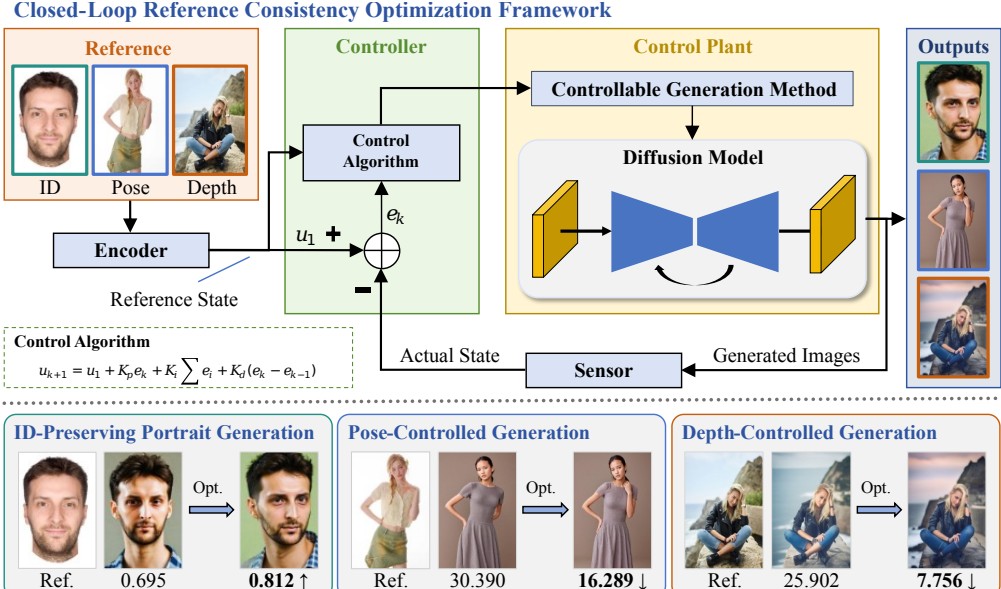

Figure 1: **Upper**: The closed-loop reference consistency optimization framework. The state of the generated image is observed and fed back via a sensor. A controller then dynamically adjusts the input to the control plant, and the deviation from the reference image state is reduced through an iterative optimization approach. **Lower**: Typical applications of the framework: (1) ID-preserving portrait generation, (2) pose-controlled generation, and (3) depth-controlled generation. The metrics are facial similarity, pose distance and depth error.

Under this formulation, current reference-consisten image generation methods resemble open-loop control systems: they inject reference conditions (e.g., image features, spatial layouts) into the generative model (the plant) in a single, forward pass without subsequent verification of the output against the desired target. Consequently, the fidelity of the result depends entirely on the model's pre-trained capabilities, lacking any mechanism for real-time error detection or correction. In contrast, closed-loop architectures dynamically adjust control actions based on real-time feedback, endowing them with superior robustness and precision. We therefore posit that **reformulating reference-consistent image generation as a closed-loop control system** enables iterative optimization of the output, thereby achieving superior reference consistency.

To materialize this vision, we propose a novel framework for reference-consistent image generation. It enables iterative inference-time optimization to achieve precise alignment with various control conditions, as shown in Figure 1. Following control principles, our system measures deviations using feature extractors as sensors, and regulates them via a controller for stable output convergence. This framework does not require any modifications to the generative model; it only needs a few lines of code addition to the script to improve the reference consistency of the generated images. To evaluate the the effectiveness and versatility of the framework, we applied this framework to three representative applications: (1) ID-preserving portrait generation, (2) pose-controlled generation, and (3) depth-controlled generation. The contributions of this work can be summarized as follows:

• To the best of our knowledge, this work is the first to integrate control theory into controllable image generation, providing a novel perspective for refining generated content. It demonstrates that controllable image generation is a control problem, where control theory can be applied for optimization.

• We propose a reference-consistent optimization framework based on a closed-loop control architecture that operates without additional model training, offers straightforward implementation, and supports diverse controllable generation methods and control conditions.

• Based on this framework, we develop simple yet effective methods to enhance reference alignment, achieving performance improvements of 12.07% for ID-preserving portrait generation, 32.64% for pose-controlled generation, and 33.49% for depth-controlled generation.

## 2 RELATED WORKS

### 2.1 CONTROLLABLE GENERATION WITH DIFFUSION MODELS

Leveraging the powerful generative capabilities of diffusion models, researchers are increasingly investigating controllable generation techniques. Among them, adapter methods have garnered considerable interest owing to their inherent flexibility and preserved generalization capacity (Mou et al., 2024; Ye et al., 2023; Zhang et al., 2023a; Rowles et al., 2024; Huang et al., 2025; Zhao et al., 2023). ControlNet++ (Li et al., 2024c) pioneers the exploration of reference consistency in controllable image generation. This method significantly enhances the spatial controllability by introducing an additional consistency loss during the training process. Despite their notable advancements, these methods still suffer from the inherent limitations of the control methods themselves, as they lack mechanisms to observe and feedback on the actual state—thus failing to achieve inference-time optimization capabilities.

Building upon prior works, we construct a closed-loop framework for controllable image generation. By continuously measuring and correcting the deviation between the generated images and the reference at inference time, this framework achieves enhanced performance.

### 2.2 REFINEMENT FOR AI-GENERATED IMAGES

In current text-to-image generation systems, cascaded architectures combining generators and refiners have been widely adopted (Podell et al., 2023; Pernias et al., 2023; Li et al., 2024b;a). This approach first uses a generator—typically a basic diffusion model—to produce a low-resolution initial image capturing core semantics and composition. A subsequent refiner then enhances visual fidelity, sharpness, and fine details. Recently, studies (Yang et al., 2024a; Mañas et al., 2024) have proposed improving image generation quality by optimizing the prompts of generative models. These approaches leverage the powerful capabilities of large language models (LLMs) to dynamically refine prompts through iterative optimization.

Unlike previous studies that primarily focus on image quality enhancement, our work specifically optimizes reference consistency through a closed-loop framework.

## 3 PRELIMINARIES

### 3.1 CLOSED-LOOP CONTROL SYSTEM

Closed-loop control system is an automated regulation system that continuously adjusts its output through real-time feedback, consisting of four key components: (1) *reference* that defines the target value of the system; (2) *controller* that acts as the decision-making unit by calculating control outputs based on the deviation (difference between reference and actual value) to minimize error; (3) *control plant* representing the equipment that requiresires regulation; and (4) *sensor* that monitors the output and feeds data back to the input. Compared to open-loop systems, closed-loop architecture demonstrates significantly improved disturbance rejection and control accuracy.

### 3.2 PID CONTROL ALGORITHM

Controllable image generation models are difficult to model using methods such as transfer functions and state-space equations, making it hard to construct closed-loop systems with LQR (Linear Quadratic Regulator) (Schaub & Konigorksi, 2023) or MPC (Model Predictive Control) (Kouvaritakis & Cannon, 2016). In contrast, PID (Proportianl Integral Derivative) (Borase et al., 2021) requires no system model, is easy to implement and tune, and is therefore selected as the controller.

The PID control algorithm is the most widely used in closed-loop control systems. It achieves precise control through the synergistic action of three components: proportional ($P$), integral ($I$), and derivative ($D$) terms. The $P$ term provides an instantaneous response proportional to the error, the $I$ term eliminates steady-state error, and the $D$ term suppresses overshoot and enhances stability. The linear combination of these three components constitutes the control output, which is expressed by Equation (1) for discrete systems.

$$u_{k+1} = K_p e_k + K_i T_s \sum_{n=0}^{k} e_n + K_d \frac{e_k - e_{k-1}}{T_s}, \tag{1}$$

where the coefficients $K_p$, $K_i$, and $K_d$ represent the gains *P*, *I*, and *D*, respectively; $u_{k+1}$ is the control signal of the $k+1$ time step; $e_k$ denotes the error between the reference and the actual output at the $k$ time step; $T_s$ is the sampling period, which refers to the fixed time interval for signal sampling in discrete control systems.

# 4 METHOD

## 4.1 FRAMEWORK OVERVIEW

Following the principles of control systems, our closed-loop optimization framework (illustrated in Figure 1), consisting of five key components: *reference*, *encoder*, *controller*, *controlled plant*, and *sensor*.

In this framework, the conditional input serves as *reference*, which defines the target value that the control system aims to achieve for the controlled plant's output. A shared feature extractor $\mathcal{F}$ is employed with dual functionality: the *encoder* analyzes the state of the reference $I^*$ to define it as the generation target, while the *sensor* monitors generated images to provide feedback. The *controller* quantifies deviations between the reference and actual states, then computes corrective signals via control algorithms $\mathcal{C}$. The generative model, integrated with a controllable generation method (denoted as $\mathcal{G}$), acts as the controlled plant to generate images based on both the corrective signal and text prompt $c$. Through closed-loop processing, this framework iteratively optimizes the reference consistency of the output images.

In the $k^{th}$ iteration, the sensor observes the actual state from the generated image $I_k$ to implement feedback. The controller first calculates the error $e_k$ between the reference state and the actual state, then employs control algorithms to compute the corrective signal $u_{k+1}$. Subsequently, the generative model leverages $u_{k+1}$ for generating the image $I_{k+1}$. This iterative closed-loop process enhances the reference consistency of the generated images, which can be formulated as Equation (2).

$$\begin{cases} u_1 = \mathcal{F}(I^*), \\ I_k = \mathcal{G}(u_k, c), \\ u_{k+1} = \mathcal{C}(u_1, \mathcal{F}(I_k)), \\ I_{k+1} = \mathcal{G}(u_{k+1}, c), \end{cases} \tag{2}$$

where $I_k$ are the generated image in the the $k^{th}$ iteration; $u_k$ is the corrective signal.

## 4.2 CONTROLLER

Dynamic systems are characterized by their output state at the current time step depending on both the instantaneous input and the output state from the previous time step—meaning such systems exhibit "state memory" of historical outputs. Conventional PID control algorithms are specifically designed for this type of dynamic system, as their core operating mechanism relies solely on the system error (i.e., the deviation between the reference and actual states) across successive time steps. In contrast, image generation systems possess the properties of a static system: their output (generated image) depends exclusively on the instantaneous input (e.g., text prompts, corrective signals) at the current iteration, with no inherent reliance on historical output states (e.g., images generated in previous iterations). This fundamental discrepancy in system characteristics renders traditional PID control algorithms fundamentally incompatible with image generation tasks. To address this issue, we propose a modified PID algorithm that incorporates the reference state at each iteration step, which is expressed in Equation (3). The sampling period corresponds to each iterative generation step, so $T_s = 1$.

$$\begin{cases} e_k = u_1 - \mathcal{F}(\mathcal{G}(u_k, c)), \\ u_{k+1} = u_1 + K_p e_k + K_i \sum_{n=0}^{k} e_n + K_d(e_k - e_{k-1}). \end{cases} \tag{3}$$

### 4.3 FRAMEWORK APPLICATIONS

**ID-preserving portrait generation**. ID-preserving portrait generation is a classic subject-driven generation task that imposes high requirements on the consistency of reference images. In this study, we constructed two closed-loop systems, with IP-Adapter-FaceID-Portrait [1] (IPA) and Kolors-IP-Adapter-FaceID-Plus [2] (KIPA) respectively employed as the control plants. InsightFace [3] is an integrated facial analysis toolkit that supports two core functions: face detection (Deng et al., 2020) and feature encoding (Deng et al., 2019). In our framework, the encoder module of InsightFace assumes a dual role—acting both as an encoder and a sensor: on one hand, it generates reference face embeddings; on the other hand, it monitors and extracts the facial embeddings of the actually generated portraits. The controller in the system calculates corrective signals by quantifying the deviation between the reference face embeddings and the actual face embeddings.

**Pose-controlled generation**. ControlNet (Zhang et al., 2023a) is the most prevalent spatial control method for image generation, compatible with various control conditions. In this work, we integrate the proposed framework with ControlNet, establishing a closed-loop optimization approach for pose-controlled generation. The OpenPose detector (Cao et al., 2019) serves as an encoder and a sensor. The controller computes corrective signals by analyzing joint position deviations between the target pose and the actual pose. Specifically, the corrective signals need to be transformed into pose maps to serve as input for ControlNet.

**Depth-controlled generation**. For the depth control task, the Midas depth estimation model (Ranftl et al., 2022) acts as an encoder and sensor. The controller computes corrective signals by evaluating pixel-wise deviations between the normalized target depth map and the actual depth map.

## 5 EXPERIMENTS

### 5.1 EXPERIMENTAL SETUP

**Configuration**. Experiments were conducted on an NVIDIA GeForce 4090 GPU. PID coefficients $K_p$, $K_i$, $K_d$ are set to 0.3, 0.05, 0.01 for ID-preserving portrait generation, and 0.05, 0.02, 0.02 for pose- and depth-controlled generation. For ID-preserving tasks, the output resolution is $1,024 \times 1,024$; for pose- and depth-controlled tasks, the shorter edge of output images is fixed at 1024 pixels, with the longer edge adaptively scaled to match the reference image's aspect ratio.

**Evaluation Dataset**. For ID-preserving portrait generation, we evaluate multi-reference and single-reference tasks. The multi-reference task uses Web100, a test set of non-celebrity social media samples, containing 130 identities (2–5 photos each). The single-reference task employs CelebA300, 300 random identities from CelebA-HQ (Karras et al., 2017). For pose controlled and depth controlled generation, we use a filtered test set from a public dataset [4], retaining only single-person photos (477 samples total).

**Metrics**. For ID-preserving portrait generation, we measure ID fidelity using facial similarity computed via the Insightface toolkit. Since embedding-based methods often encode facial pose and expression into representations—leading to insufficient variation in generated facial regions—we propose a "structure diversity" metric: facial landmarks (Deng et al., 2018) are detected in all generated and reference images, and the average pairwise distance between these landmarks is calculated (larger values indicate higher diversity). For pose-controlled generation, body joints in both image sets are detected via OpenPose (Cao et al., 2019), with mean Euclidean distance computed across all matched keypoints. For depth-controlled generation, depth consistency is assessed via pixel-wise deviation: depth values are extracted using the MiDaS estimator (Ranftl et al., 2022), followed by calculation of pixel-wise mean absolute error (MAE). Additionally, CLIP-I (Radford et al., 2021) and DINO (Caron et al., 2021) evaluate semantic similarity between generated and reference images, while Q-Align (Wu et al., 2023) assesses generation quality.

---

[1] https://huggingface.co/h94/IP-Adapter-FaceID

[2] https://huggingface.co/Kwai-Kolors/Kolors-IP-Adapter-FaceID-Plus

[3] https://github.com/deepinsight/insightface

[4] https://huggingface.co/datasets/raulc0399/open_pose_controlnet

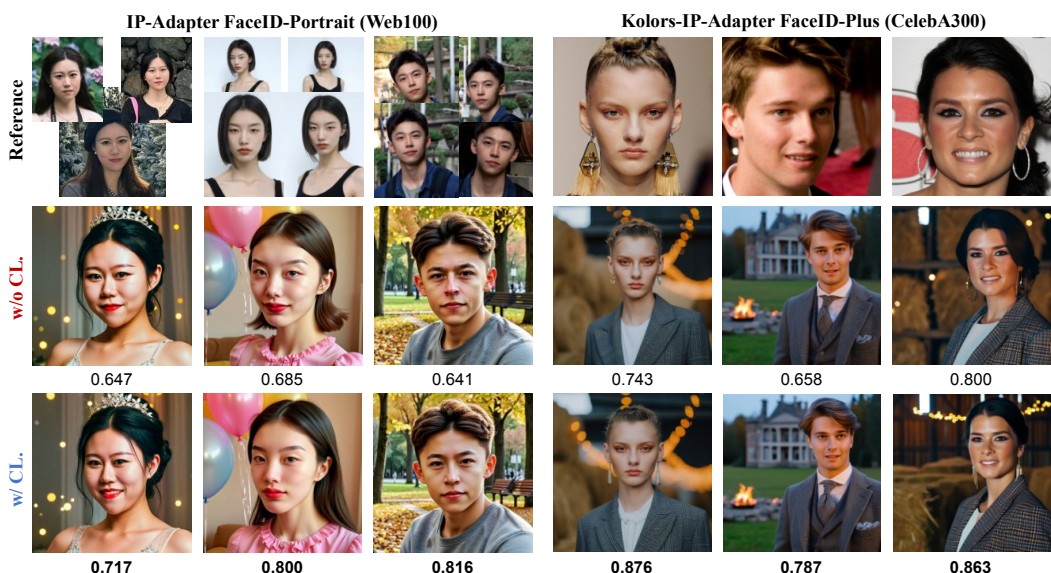

Figure 2: Qualitative comparison on ID-preserving portrait generation. **First row**: reference images. **Second row**: generated images without closed-loop optimization. **Third row**: generated images with closed-loop optimization. **Left**: results on Web100. **Right**: results on CelebA300. The average facial similarity (↑) to the reference image is shown below.

## 5.2 QUALITATIVE EVALUATION

The proposed closed-loop optimization framework demonstrates superior identity consistency compared to non-optimized generation methods, as shown in Figure 2. The optimized portraits exhibit a higher degree of similarity to the reference images in terms of facial features, facial structure, skin texture, and other relevant aspects. Furthermore, we conducted multi-round generation experiment, where the image generated in the previous round was used as the reference image for the next round. Under this working condition, a decrease in portrait similarity is inevitable. As shown in Figure 3, the use of closed-loop optimization has effectively reduced the rate of decline in facial similarity, making the generation process more robust.

Figures 4 and 5 illustrate that our framework substantially reduces pose distance and depth error, respectively—further validating its effectiveness and versatility. Enabled by closed-loop optimization, the pose and depth of the generated images are more aligned with the corresponding references.

## 5.3 QUANTITATIVE EVALUATION

We computed the metric during the closed-loop optimization process (total iterations: 20 for ID-preserving generation, 15 for pose-controlled and depth-controlled generation), as shown in Figure 6. The results show that performance improves significantly during the early optimization stages and then converges.

Experimental results quantitatively demonstrate the effectiveness and versatility of the closed-loop optimization framework. For ID-preserving generation, both IPA and KIPA have enhanced facial similarity through integration with the proposed closed-loop framework, as shown in Table 1. Specifically, IPA achieves 9.75% improvement on the CelebA300 dataset and 8.78% improvement on the Web100 dataset, while KIPA achieves 10.13% on CelebA300 and 12.07% on Web100. Both methods retain performance at the original level across other evaluation metrics. Additionally, our results are also competitive when compared to mainstream methods (Wang et al., 2024; Li et al., 2024d; Guo et al., 2024).

The proposed method yields substantial performance gains, with 32.64% in pose consistency and 33.49% in depth consistency. This significant enhancements are achieved while maintaining high generation quality.

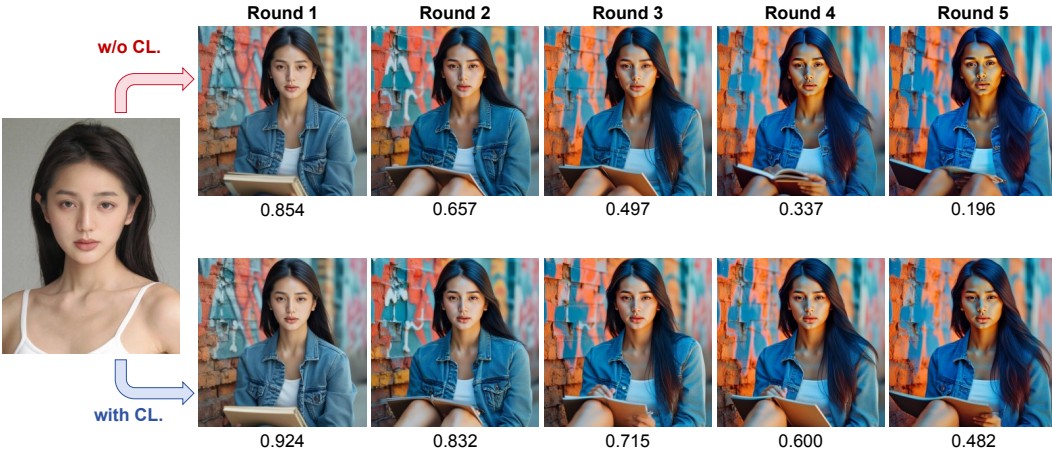

Figure 3: Multi-round generation experiment, where the image generated in the previous round was used as the reference image for the next round. The use of closed-loop optimization has effectively reduced the rate of decline in facial similarity. Facial similarity (↑) to the reference image is shown below.

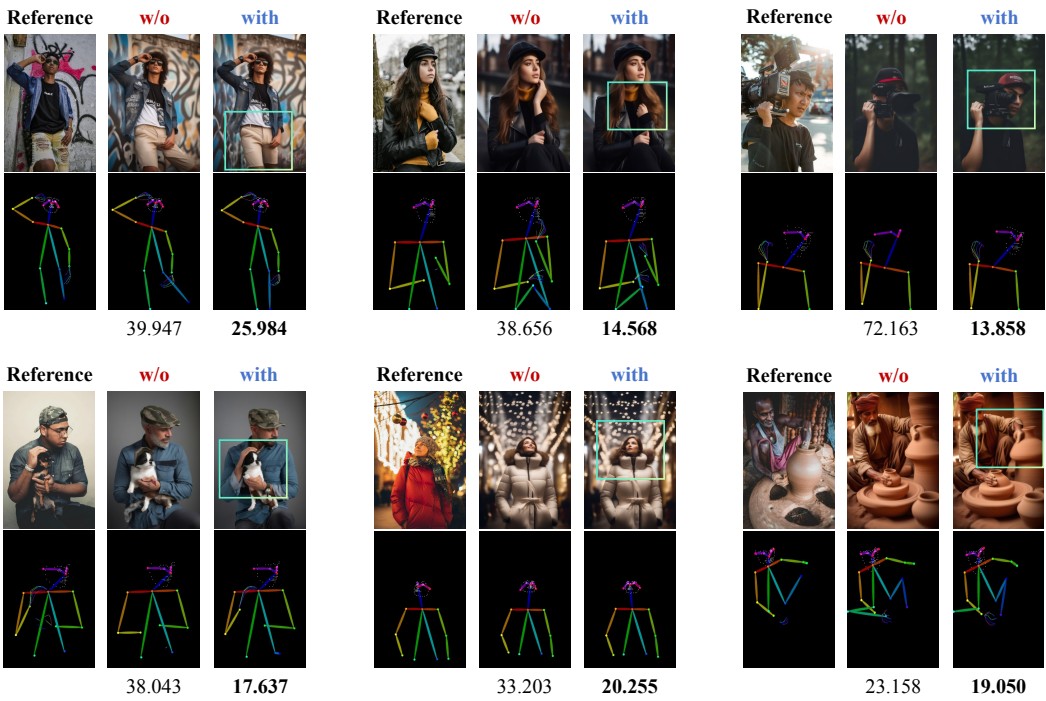

Figure 4: Qualitative comparison on pose-controlled generation. Each cell contains a reference image, a generated image without closed-loop optimization, and a generated image with closed-loop optimization, along with their corresponding pose maps. The average pose distance (↓) to the reference pose is shown below. The significant differences are highlighted.

## 5.4 ABLATION STUDY

**Impact of PID components**. We conducted ablation studies on the influence of the $P$, $I$, and $D$ components in the optimization methods. For ID-preserving portrait generation, we tested the IPA model on the Web100 dataset; for pose and depth-controlled generation, we used the entire test set. The results are shown in Table 3. For ID-preserving portrait generation, the $P$ and $I$ combination effectively improve the facial similarity while maintaining generation quality, achieved the best performance. For pose-controlled generation, all three PID components ($P$, $I$, $D$) contributed

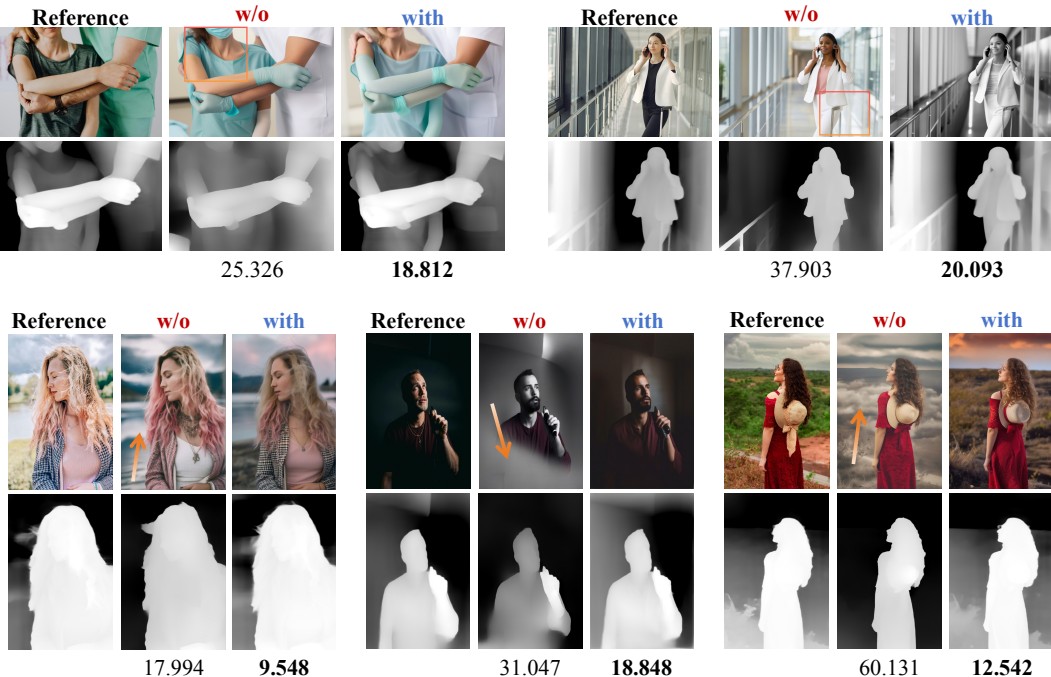

Figure 5: Qualitative comparison on depth-controlled generation. Each cell contains a reference image, a generated image without closed-loop optimization, and a generated image with closed-loop optimization, along with their corresponding depth maps. The average depth error ($\downarrow$) to the reference image is shown below. The significant differences are highlighted.

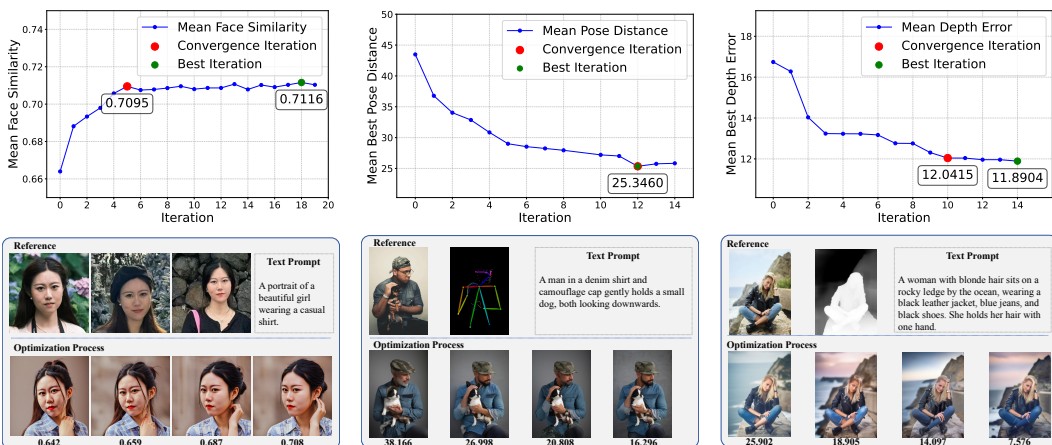

Figure 6: **Upper**: Process of closed-loop optimization on ID-preserving portrait generation, pose-controlled generation and depth-controlled generation. **Lower**: Visualization examples of closed-loop optimization process.

positively by reducing the distance between generated and reference poses. For depth-controlled generation, the combination of *P* and *D* combination achieved optimal results. The results demonstrate that the proposed closed-loop optimization method can effectively enhance reference consistency across various control conditions.

**Effectiveness across different text levels**. We investigate the impact of text prompts with varying complexity levels—no prompt, simple, moderate, and complex. We randomly selected 10 data samples from each of the Web100 dataset, the pose-controlled generation dataset, and the depth-controlled generation dataset. As shown in Table 4, the proposed closed-loop optimization framework

Table 1: Quantitative evaluation on ID-preserving portrait generation. **Bold**: the best results, and underline: the suboptimal results. [Key: CL.: closed-loop optimization; Sim.: facial similarity, Div.: structure diversity.]

| Method | CelebA300 | | | | | Web100 | | | | |
|---|---|---|---|---|---|---|---|---|---|---|
| | Sim. ↑ | CLIP-I ↑ | DINO ↑ | Q-Align ↑ | Div. ↑ | Sim. ↑ | CLIP-I ↑ | DINO ↑ | Q-Align ↑ | Div. ↑ |
| PuLID | 0.532 | 0.734 | 0.356 | 4.969 | 0.063 | 0.596 | 0.797 | 0.412 | 4.964 | 0.121 |
| Photomaker v2 | 0.556 | 0.764 | 0.360 | 4.967 | 0.085 | 0.600 | **0.819** | 0.435 | 4.928 | 0.131 |
| InstantID | 0.819 | **0.788** | **0.599** | 4.199 | 0.023 | 0.723 | 0.802 | 0.480 | 4.035 | 0.101 |
| IPA | 0.708 | 0.756 | 0.529 | 4.946 | **0.114** | 0.672 | 0.798 | 0.492 | 4.973 | 0.149 |
| IPA + CL. (ours) | 0.777 | 0.774 | 0.532 | 4.950 | 0.108 | 0.731 | 0.808 | 0.486 | **4.977** | **0.151** |
| KIPA | 0.750 | 0.767 | 0.533 | **4.977** | 0.043 | 0.671 | 0.813 | 0.496 | 4.951 | 0.111 |
| KIPA + CL. (ours) | **0.826** | 0.773 | 0.535 | 4.971 | 0.042 | **0.752** | 0.816 | **0.497** | 4.947 | 0.107 |

Table 2: Quantitative evaluation on pose-controlled and depth-controlled generation. **Bold**: the best results. [Key: CL.: closed-loop optimization; Dist.: pose distance, Err.: depth error.]

| Method | Pose-Controlled Generation | | | | Depth-Controlled Generation | | | |
|---|---|---|---|---|---|---|---|---|
| | Dist. ↓ | DINO ↑ | CLIP-I ↑ | Q-Align ↑ | Err. ↓ | DINO ↑ | CLIP-I ↑ | Q-Align ↑ |
| ControlNet | 39.193 | 0.6297 | 0.7891 | 4.936 | 16.983 | 0.781 | 0.840 | 4.956 |
| ControlNet + CL. (ours) | **26.401** | **0.6300** | **0.7887** | **4.940** | **11.295** | **0.785** | **0.843** | **4.963** |

consistently improves performance across all prompt complexity levels. This validates that the closed-loop mechanism effectively mitigates ambiguities introduced by textual instructions, ensuring robust generation quality.

**Effectiveness across different random seeds**. Random seeds significantly impact image generation quality. We validated the influence of random seeds, conducting statistics on generation results across multiple random seeds as shown in Table 5. We randomly select 10 data samples from each testset. The results demonstrate that the closed-loop optimization framework improves the mean value of performance metrics while reducing their standard deviation, indicating that this method enhances the robustness of controllable image generation systems. Specifically, the increase in the mean value of performance metrics indicates that our method can effectively enhance the consistency of generated images, while the reduction in the standard deviation reflects that the method mitigates performance fluctuations caused by randomness.

Table 3: Ablation study with different PID component combinations. **Bold**: the best results, and underline: the suboptimal results. [Key: Sim.: facial similarity, Dist.: pose distance, Err.: depth error.]

| Method | P | I | D | Identity | | Pose | | Depth | |
|---|---|---|---|---|---|---|---|---|---|
| | | | | Sim. ↓ | FID ↓ | Dist. ↓ | FID ↓ | Err. ↓ | FID ↓ |
| W/o Closed loop | ✗ | ✗ | ✗ | 0.664 | 131.7 | 39.193 | 116.7 | 16.983 | 102.5 |
| | ✓ | ✗ | ✗ | 0.686 | 131.5 | 29.417 | 116.5 | 11.316 | 102.3 |
| Closed loop | ✓ | ✓ | ✗ | **0.7243** | **129.7** | 26.475 | 117.5 | 11.767 | **101.3** |
| (ours) | ✓ | ✗ | ✓ | 0.687 | 132.0 | 28.645 | 116.4 | **11.295** | 101.7 |
| | ✓ | ✓ | ✓ | 0.7236 | 129.9 | **26.401** | **116.0** | 11.798 | 102.3 |

Table 4: Ablation study on text prompts with different complexity levels. (Metrics: ID-preserving portrait generation: facial similarity; pose-controlled generation: pose distance; depth-controlled generation: depth error).

| Text Prompt Complexity Level | Identity | | Pose | | Depth | |
|---|---|---|---|---|---|---|
| | W/o CL. | With CL. | W/o CL. | With CL. | W/o CL. | With CL. |
| No prompt | 0.651 | **0.707** | 29.952 | **22.066** | 22.598 | **14.553** |
| Simple | 0.592 | **0.667** | 26.240 | **21.810** | 21.638 | **12.610** |
| Moderate | 0.596 | **0.659** | 27.064 | **22.491** | 22.970 | **13.269** |
| Complex | 0.593 | **0.666** | 26.677 | **21.324** | 21.902 | **12.576** |

Table 5: Ablation study on random seeds. (Metrics: ID-preserving portrait generation: facial similarity; pose-controlled generation: pose distance; depth-controlled generation: depth error).

| | Identity | | Pose | | Depth | |
|---|---|---|---|---|---|---|
| | Mean ↑ | Std. ↓ | Mean ↓ | Std. ↓ | Mean ↓ | Std. ↓ |
| W/o CL. | 0.658 | 0.021 | 23.618 | 9.050 | 20.808 | 5.348 |
| With CL. | **0.706** | **0.009** | **18.220** | **5.629** | **10.942** | **1.975** |

## 5.5 ANALYSIS OF COMPUTATIONAL EFFICIENCY

We conducted a computational efficiency analysis for identity-preserving portrait generation and pose-controlled generation (pose control and depth control are similar in terms of efficiency). The computing resource usage during the closed-loop optimization process of the two tasks is shown in the Figure 7. For identity-preserving portrait generation, the average iteration time per run is 4.2 seconds, with a peak video memory usage of 15,286 MB and a peak CPU utilization rate of 76.6%. For pose-controlled generation, the average iteration time per run is 11.6 seconds, accompanied by a peak video memory usage of 14,148 MB and a peak CPU utilization rate of 49.7%. The total generation time depends on the setting of the maximum number of iterations, and the time and computing resource consumption of each iteration are basically consistent with those of a single round generation.

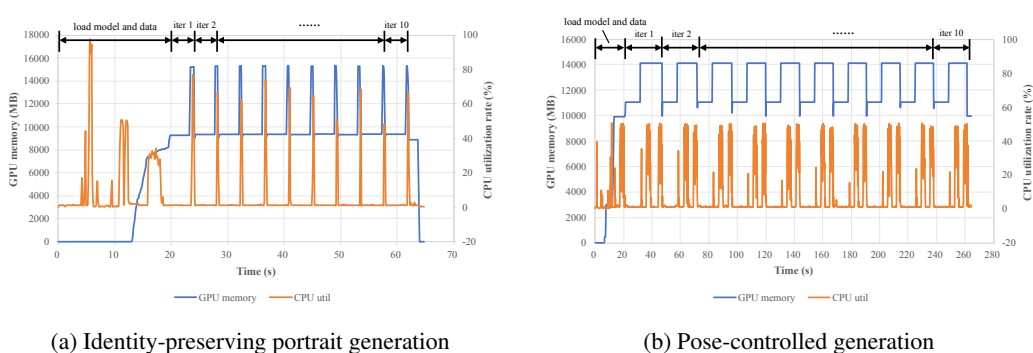

(a) Identity-preserving portrait generation          (b) Pose-controlled generation

Figure 7: The computing resource usage during the closed-loop optimization process.

## 6 CONCLUSION

This study proposes a inference-time iterative optimization framework, which introduces automatic control theory into the field of image generation and significantly improves the reference consistency through a closed-loop feedback architecture. The proposed optimization framework has been extensively validated across multiple tasks, demonstrating its effectiveness and versatility. This work not only presents a solution for reference-consistent image generation but also puts forward a novel perspective: controllable image generation can be framed as a control problem, where control theory is applicable for performance optimization.

**Limitations and future works**:

• This work employs a PID algorithm in the controller, with parameters relying on manual tuning, which increases the workload in practical applications. Future work will adopt advanced control algorithms.

• While iterative optimization improves reference consistency of generated images, its multi-round nature leads to higher computational resource and time consumption. Future work will focus on faster convergence.

STATEMENT

**ETHICS STATEMENT**: The research conducted in the paper conforms, in every respect, with the ICLR Code of Ethics.

**REPRODUCIBILITY STATEMENT**: We have provided implementation details in sec. 5. We will also release all the code.

**LLM USAGE STATEMENT**: Large Language Models (LLMs) were used solely for polishing writing. They did not contribute to the research content or scientific findings of this work.

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

## A  PID COEFFICIENTS TUNING TECHNIQUES

The tuning of PID coefficients relies on manual adjustment. We have summarized the following tuning techniques.

$K_p$ **tuning**:

- Start with $K_p = 0$ and gradually increase until the system exhibits oscillations.

- Reduce $K_p$ to 50% of the oscillatory value (or adjust within $0.4$–$0.8$ critical gain for stability).

$K_i$ **tuning**:

- With $K_p$ fixed, start from $K_i = 0$ and gradually increase $K_i$ until steady-state error is eliminated.

- Anti-Windup Measures:

  - Integral clamping: Limit the integral term to prevent excessive accumulation.

  - Integral separation: Disable integration when error exceeds a threshold.

$K_d$ **tuning**: Gradually increase $K_d$ to suppress overshoot and improve response speed.

## B  PSEUDOCODE

This algorithm improves generated images' reference consistency with low code modification costs — only a few core lines need adding to a basic generative model's original logic (see Algorithm 1) to implement closed-loop optimization. Its core logic has three stages, with added code focused on "error calculation" and "PID adjustment":

- Initialization (2–3 new lines) : Add code to extract the reference image's feature signal $s_0^*$ via sensor $\mathcal{F}$, and initialize error $\delta_0$ and integral term $\Sigma_\delta$ — no model reconstruction needed.

- Iterative optimization (4–5 new lines) : After the model outputs image $I$, add code to extract $I$'s features/calculate error $\delta$ (via $\mathcal{F}$), update input signal $s^*$ with PID, and refresh historical error $\delta_0$; reuse the original loop framework.

- Result output (no new lines) : Keep the original image output logic to get results with features highly consistent with the reference.

Overall, no large-scale modifications to the generative model $\mathcal{G}$ or sensor $\mathcal{F}$ are required. Fewer than 10 core lines (for closed-loop feedback and PID) suffice to significantly boost reference consistency, meeting the "small modifications, great results" goal.

---

**Algorithm 1** Closed-loop optimization of reference-consistent image generation

---

**Input:** Reference image $I^*$, PID coefficients ($K_p$, $K_i$, $K_d$), generative model $\mathcal{G}$, sensor $\mathcal{F}$, max iteraion $N$.
**Output:** Generated image $I$.
1  $\delta_0 \leftarrow \mathbf{0}, \Sigma_\delta \leftarrow \mathbf{0}, n \leftarrow 0$        # Initial error, initial integral of errors, initial counter
2  $s_0^* \leftarrow \mathcal{F}(I^*)$        # Initial reference state
3  $s^* \leftarrow s_0^*$
4  **while** $n \leq N$ **do**
5     $I \leftarrow \mathcal{G}(s^*)$        # Generate image
6     $s \leftarrow \mathcal{F}(I)$        # Measure actual state from generated image
7     $\delta \leftarrow s^* - s$        # Compute error
8     $\Sigma_\delta \leftarrow \Sigma_\delta + \delta$        # Update integral
9     $s^* \leftarrow s_0^* + K_p \delta + K_i \Sigma_\delta + K_d(\delta - \delta_0)$        # PID control law
10     $\delta_0 \leftarrow \delta$        # Save current error
11     $n \leftarrow n + 1$        # Increment counter
12  **end**
13  **return** $I$

---

## C ANALYSIS OF COEFFICIENT STABILITY

We use IPA as the baseline and randomly select 25 groups of data from the Web100 dataset to test the stability of the three coefficients ($K_p$, $K_i$, $K_d$) in the closed-loop optimization method. Specifically, one coefficient is adjusted while the other two are fixed, and the changes in the average facial similarity are observed. The base coefficient are set as $K_p = 0.3$, $K_i = 0.05$, and $K_d = 0.01$. As shown in Figure A-1, the results indicate that all three coefficients are effective within a wide range.

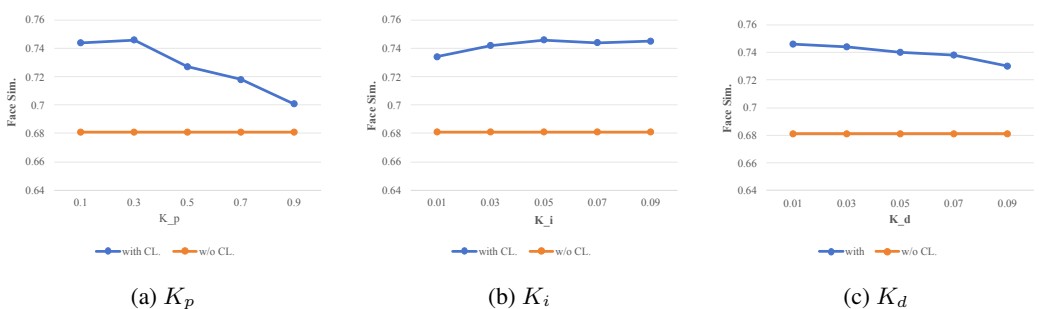

(a) $K_p$                        (b) $K_i$                        (c) $K_d$

Figure A-1: Stability Analysis of $K_p$, $K_i$ and $K_d$.

## D ADDITIONAL RESULTS

The additional visualization results of identity-preserving portrait generation, pose-controlled generation, and depth-control generation are shown in Figure A-2, A-3, A-4 and A-5, respectively. The results across all three tasks collectively demonstrate the effectiveness of the closed-loop framework in improving reference consistency. Qualitative comparisons between methods for ID-preserving portrait generation are shown in Figure A-6.The additional multi-round generation result are shown in Figure A-7.

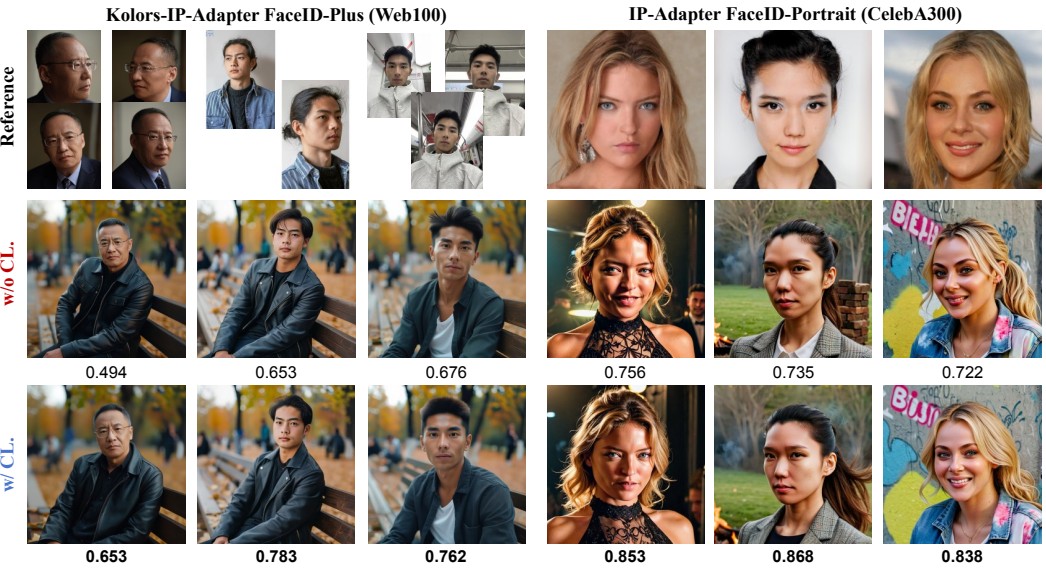

Figure A-2: Qualitative comparison on ID-preserving portrait generation.

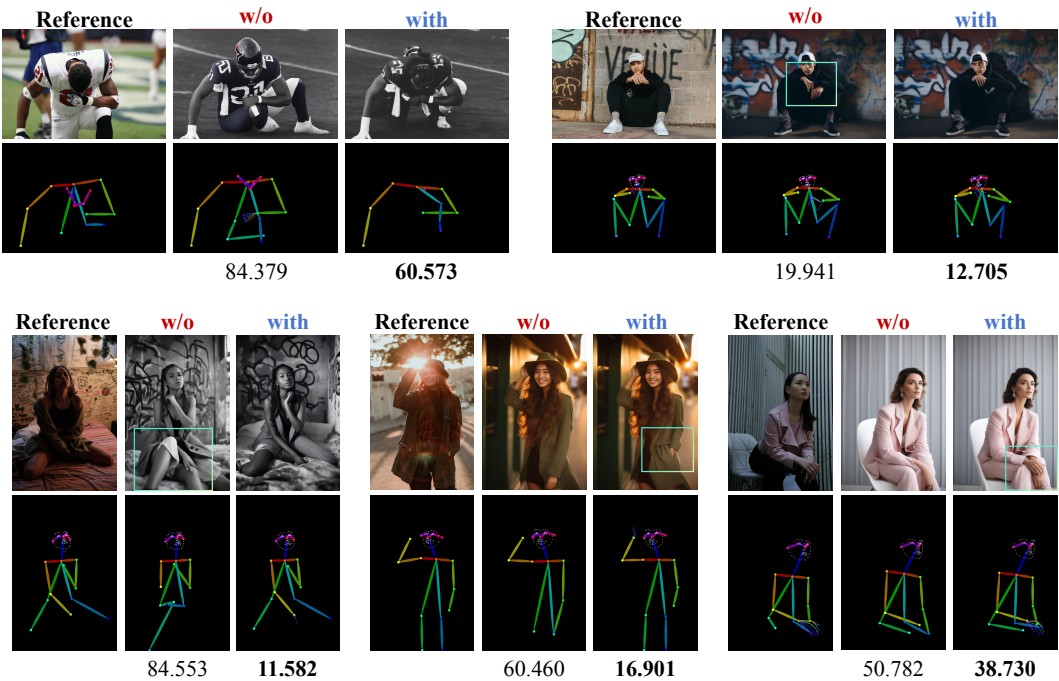

Figure A-3: Qualitative comparison on pose-controlled generation.

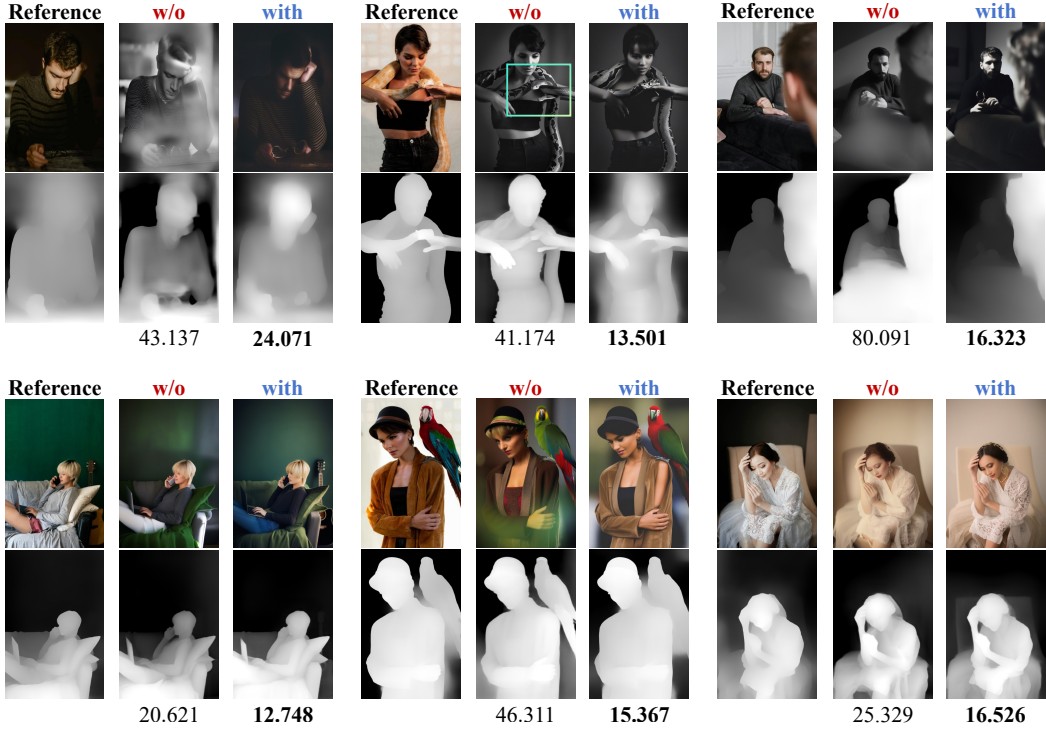

Figure A-4: Qualitative comparison on depth-controlled generation.

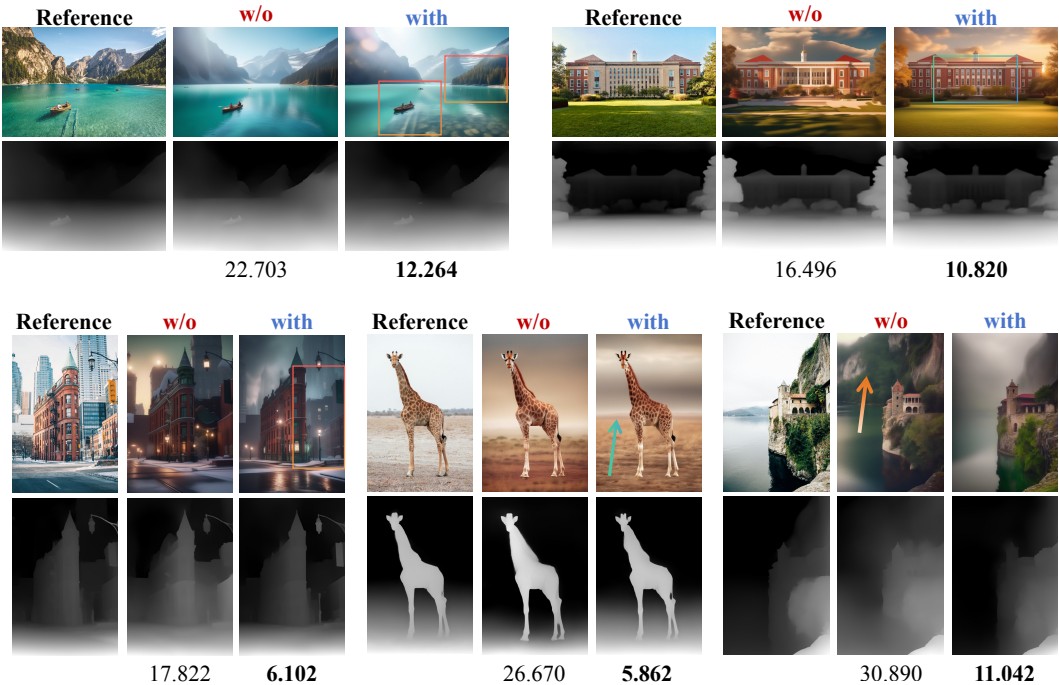

Figure A-5: Qualitative comparison on depth-controlled generation (non-human samples).

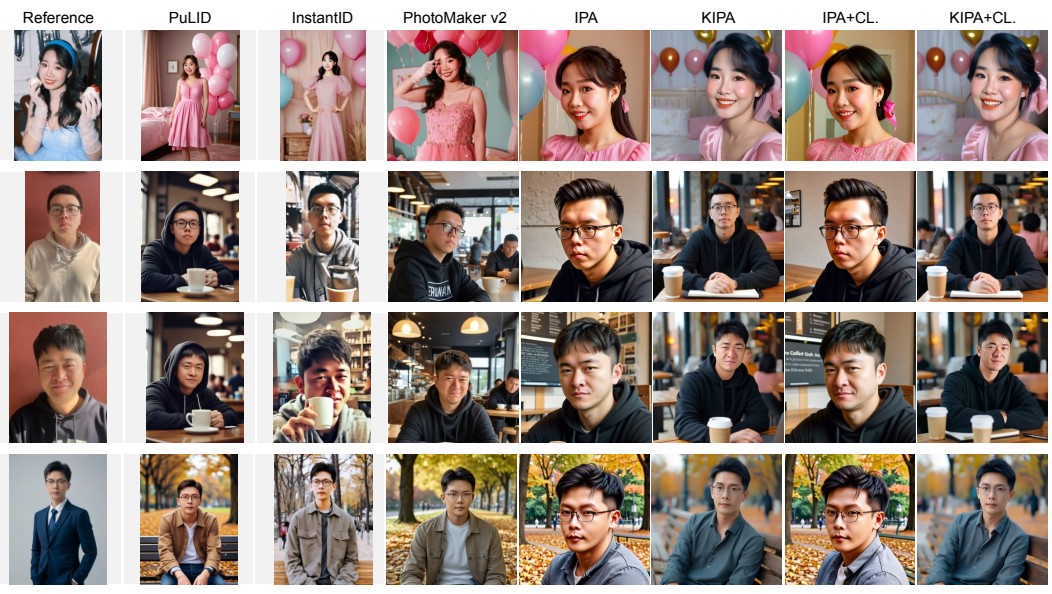

Figure A-6: Qualitative comparisons between mainstream methods for ID-preserving portrait generation.

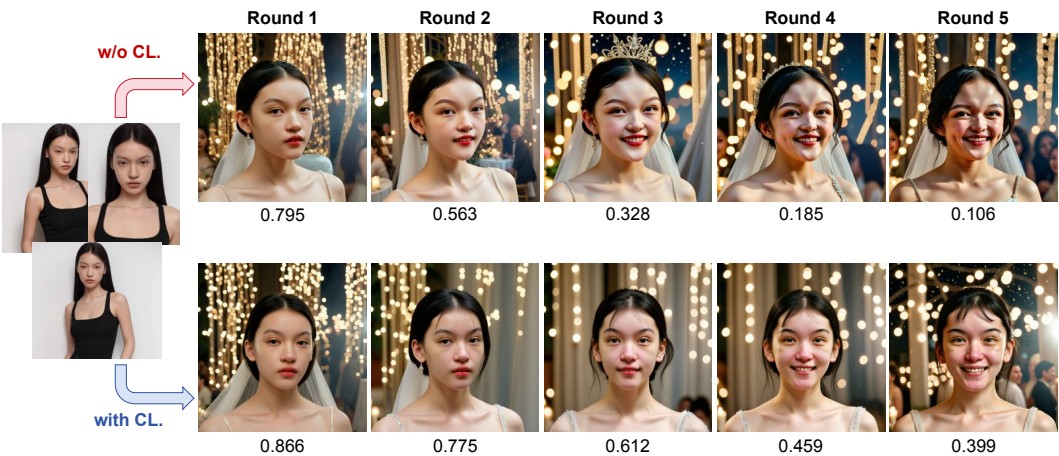

Figure A-7: Multi-round generation experiment, where the image generated in the previous round was used as the reference image for the next round.

