# OpenReview forum: "CL-Gen: An Inference-Time Iterative Optimization Framework for Reference-Consistent Image Generation Based on Closed-Loop Control"
_ICLR.cc/2026/Conference — Submitted to ICLR 2026_

### Official Review · Reviewer_QLbv · 2025-10-26

**Soundness:** 2
**Presentation:** 1
**Contribution:** 2
**Rating:** 2
**Confidence:** 3

**Summary:**

This paper addresses the task of controllable image generation with a focus on improving reference consistency between generated images and user-provided conditions. The authors propose a closed-loop, iterative optimization framework inspired by control theory, using a PID controller to dynamically adjust the generation model's inputs based on feedback during inference. The method is compatible with various controllable generation approaches and can be easily implemented without additional training. Experimental results demonstrate significant improvements in identity preservation, pose control, and depth control, highlighting the method's effectiveness and generalizability.

**Strengths:**

- The paper addresses an important problem with clearly articulated motivation.
- Experimental results demonstrate that the proposed method outperforms existing approaches.

**Weaknesses:**

- The paper appears to be somewhat over-packaged. Its core idea—a simple iterative feedback mechanism introduced into conditional image generation—is straightforward, yet the authors emphasize the concept of closed-loop systems from control theory, which seems unnecessary. The use of closed-loop terminology primarily serves to label different stages of the generation process, without establishing a substantive connection between the two.
- The introduction to closed-loop theory is insufficiently detailed. Most of the target readers are likely unfamiliar with this theory, yet the explanation is overly brief and may cause confusion. For example, it is unclear what u and v represent in Equation 1, what their subscripts denote, what the sampling period is, why it is needed, and what the meaning of Equation 1 is.
- In Section 4.1, the authors claim that their approach consists of five core components, but "reference" is not explained immediately afterwards. In subsequent text, "reference" appears to be the conditional input; however, it is unclear if this should be considered a separate component of the method.
- Equation 3 is presented as an improvement over Equation 1, but the rationale behind this modification is not clearly explained.
- Although the authors claim their method requires only a few lines of code to implement, the paper does not provide any concrete code examples, despite the inclusion of pseudocode in Appendix B. Intuitively, it seems unlikely that the proposed approach can be implemented by modifying just a few lines of code.
- The paper lacks any analysis or comparison of computational efficiency.

**Questions:**

See Weaknesses.

---

> ### Author Response · Authors · 2025-11-20
> **Reponse to Reviewer QLbv (Part1)**
>
> We would like to thank you for taking the time to review our work. Your questions and comments are precious. We value each of them and have provided detailed responses below：
>
> **W1:  Explanation on the necessity of emphasizing closed-loop control**
>
> Thank you for your constructive comments on the conceptual presentation and core value of the paper. We fully understand your concerns regarding the use of the "closed-loop system" concept and elaborate on the necessity and substantive significance of emphasizing this concept as follows:
>
> The reason we emphasize the closed-loop system concept from control theory is not to "overpackage the paper" or merely attach labels, but stems from in-depth reflections on the essence of controllable image generation systems. We argue that controllable image generation and traditional control systems (such as motor control and temperature control) share an isomorphic core logic — this isomorphism serves as the theoretical foundation of our method, rather than a superficial analogy.
>
> Both systems include complete core closed-loop components: The "reference signal (target value) - controller - controlled object - sensor (feedback detection)" chain in traditional control systems has clear corresponding implementations in our method: the conditional input acts as the "reference signal", the PID algorithm as the "controller", the image generation model as the "controlled object", and the error detection modules based on InsightFace, OpenPose, and MiDaS as the "sensors". This component-level correspondence is not an arbitrary division but the core basis for designing our iterative optimization process. Precisely based on this correspondence, we can migrate mature PID control logic to image generation tasks, realizing a closed-loop cycle of "error detection - dynamic adjustment - optimized output".
>
> Emphasizing the closed-loop concept is the core support for methodological innovation: The innovation of this paper does not lie in "simple iterative feedback", but in the first explicit treatment of the controllable image generation system as a closed-loop system that can be optimized via control theory. Based on this, we systematically introduce the PID controller to address the "reference consistency deviation" problem in the generation process.
>
> **W2 & W3: Supplement the Preliminary section and clarify the Method section**
>
> Following your suggestion, we have added an introduction to the basic concept of closed-loop control in the Preliminary section (Sec. 3.1), and refined the description in the Method section (Sec. 4.1) regarding how closed-loop control integrates with the generative model.
>
> In Sec. 3.1, we supplement the introduction to the composition of the closed-loop control system and the functions of its key components.
>
> In Sec. 4.1, we elaborate on the specific workflow of the iterative process. In the proposed framework, the conditional input serves as reference, which defines the target value that the control system aims to achieve for the controlled plant’s output. In one iteration, the sensor first observes the previously generated image and extracts its feature representation. The controller calculates a new input based on the deviation between the reference and the actual features. The generative model, acting as the control plant, responds to this input and generates a new image. Through multiple iterations, the features of the generated images are brought closer to the reference.
>
> Additionally, we have supplemented the explanations of the symbols used in the formulas to enhance the readability of the mathematical expressions.

---

> ### Author Response · Authors · 2025-11-20
> **Reponse to Reviewer QLbv (Part2)**
>
> **W4:  Explanation on the he rationale for revising Equation 1 to Equation 3**
>
> Thank you for your concern regarding the improved PID algorithm. The following is an explanation of the rationale for this modification.
>
> A core feature of dynamic systems is that their output state at the current time step depends not only on the instantaneous input but also on the output state from the previous time step—meaning such systems exhibit "state memory" of historical outputs. Taking a typical motor speed control system as an example: if the target is to stabilize the motor speed at 1500 rpm, the motor's current actual speed (output state) depends not only on the current input signal but also directly on the speed from the previous second. For instance, if the speed reached 1480 rpm in the previous second, only a slight increase in voltage is needed to approach the target, rather than a significant voltage boost required when the initial speed is 0 rpm. Conventional PID control algorithms are specifically designed for such dynamic systems, as their core operating mechanism (proportional-integral-derivative adjustment) relies entirely on system errors (i.e., the deviation between the reference state and the actual state) across successive time steps. They dynamically adjust the input by accumulating historical errors and predicting error variation trends to achieve stable control.
>
> In contrast, image generation systems exhibit the properties of static systems, which stand in sharp contrast to the dynamic characteristics of motor control systems: their output (generated image) is uniquely determined solely by the instantaneous input at the current iteration step (e.g., text prompts, reference image features, corrective signals), and inherently does not depend on any historical output states (including images generated in previous iterations). Specifically, in the task of "generating a smiling expression based on a reference face," if the input parameters are fixed as "reference face features + smiling semantic prompt," each generation process constitutes an independent mapping from input to output. Regardless of whether the images output in previous iterations have expression distortion issues, the new generation result is driven solely by the current input signal. It neither inherits the reasonable facial contour features from historical outputs nor avoids expression generation defects based on historical errors. This "history-independent" static property is completely contrary to the core mechanism of conventional PID control algorithms, which "rely on error accumulation across successive time steps," resulting in an inherent incompatibility between the two. Therefore, Equation 3 needs to incorporate the initial control signal $u_1$ based on Equation 1. Additionally, for the image generation system, the sampling period corresponds to each iteration, so $T_s = 1$.
>
> **W5:  Supplementation of source code**
>
> We have organized the code for the original IP-Adapter-FaceID-portrait and the version integrated with closed-loop optimization, which will be included in the supplementary materials. Through comparison, it is found that the closed-loop optimization only adds 12 lines of code. The rest of the code will be released upon publication.
>
> **W6: Addition of computational efficiency analysis**
>
> Following your suggestions, we have added an analysis of computational efficiency in Section 5.5. Specifically, we conducted a computational efficiency analysis for identity-preserving portrait generation and pose-controlled generation (pose control and depth control are similar in terms of efficiency). The computing resource usage during the closed-loop optimization process of the two tasks is shown in the Figure 7. For identity-preserving portrait generation, the average iteration time per run is 4.2 seconds, with a peak video memory usage of 15,286 MB and a peak CPU utilization rate of 76.6%. For pose-controlled generation, the average iteration time per run is 11.6 seconds, accompanied by a peak video memory usage of 14,148 MB and a peak CPU utilization rate of 49.7%. The total generation time depends on the setting of the maximum number of iterations, and the time and computing resource consumption of each iteration are basically consistent with those of a single round generation.
>
> Once again, we sincerely express our best gratitude for your effort and time!
>
> Best wishes,
>
> Team of 2798

---

### Official Review · Reviewer_3epi · 2025-10-29

**Soundness:** 3
**Presentation:** 2
**Contribution:** 3
**Rating:** 4
**Confidence:** 3

**Summary:**

The paper presents an interesting algorithm based on the idea of closed loop control for reference-consistent image generation. The proposedl algorithm is well motivated. It contains five components: reference, encoder, controller, controlled plant, and sensor. Based on the proposed algorithm, it obtains reasonable results on three different applications like ID-preserving portrait generation, Pose-controlled generation, and Depth-controlled generation.

**Strengths:**

*  Using the idea of PID control algorithm for image generation is novel. The closed loop system can well improve the generation consistency.

* The proposed algorithm can be applied to three different tasks with reasonable experimental results like ID-preserving portrait generation, Pose-controlled generation, and Depth-controlled generation.

* The reproduce of the paper should be easy as the paper provide sufficient implementation details in the paper.

**Weaknesses:**

* The experiments should involve more state-of-the-art algorithms for comparison. For example,  in the ID-preserving portrait generation experiments, it should include the comparisons with [R1] and [R2].

[R1] PuLID: Pure and Lightning ID Customization via Contrastive Alignment
https://arxiv.org/pdf/2404.16022

[R2] InfiniteYou: Flexible Photo Recrafting While Preserving Your Identity
https://arxiv.org/pdf/2503.16418

* For the ID similarity evaluation, why not use the metric based on face recognition feature like arcface, which may be a more robust metric for ID preserving evaluation. The current definition of "facial similarity" seems to be a bit weird.

* In the PID control algorithm, how to ensure the math equations corresponding the target value? For example, how to validate the value of e_k corresponding to the physical value of the error of the model outuput against the reference?

* As the algorithm involves multiple rounds of computation, the comparison with the baseline may not be fair.

* One minor suggestion: there should provide a reference for the PID when dicussing it in Section 3.

**Questions:**

Please mainly address the questions in the weakness section. More specifically, I would have more concerns on the limited experimental evaluations.

---

> ### Author Response · Authors · 2025-11-20
> **Reponse to Reviewer 3epi (Part1)**
>
> We would like to thank you for taking the time to review our work. Your questions and comments are precious. We value each of them and have provided detailed responses below：
>
> **W1: Addition of SOTA comparison**
>
> Thank you for your valuable suggestions.
>
> The primary goal of this paper is not to outperform state-of-the-art (SOTA) methods on specific tasks, but to explore whether controllable image generation can be formulated as a control problem and optimized by applying control theory. Taking three tasks—ID-preserving portrait generation, pose-controlled generation, and deep-controlled generation—as examples, we verify our hypothesis by constructing controllable image generation as a typical closed-loop control system. This system forces the features of generated images to converge toward conditional inputs, thereby achieving improved reference consistency.
>
> Your suggestions are highly valuable. Comparing with the latest SOTA methods will help demonstrate the advantages of our approach. We have added a comparison with PuLID. Quantitative evaluation results are included in Table 1, and qualitative comparisons of mainstream methods (Figure A-6), which cover PuLID, have been added in the Appendix.
>
> We did not include InfiniteYou in the comparisons, with the core reason being to ensure the fairness of the evaluation. Specifically, InfiniteYou adopts FLUX (based on the DiT architecture) as its base model, whereas our CL-GEN method is constructed via closed-loop optimization on top of IP-Adapter-FaceID-portrait and Kolors-IP-Adapter-FaceID-Plus—both of these models, along with all other comparative methods in this paper, use SDXL as their base model. Due to the significant differences in inherent generation capabilities between FLUX and SDXL, a direct comparison would conflate the performance contributions of the base models themselves with the actual effectiveness of the proposed technical innovations, failing to accurately reflect the core advantages of our closed-loop framework. To guarantee the rigor of the evaluation, we focused on comparisons among SDXL-based methods.
>
> **W2: Explanation on facial similarity metric**
>
> Thank you for your valuable comment regarding the facial similarity evaluation metric. We sincerely apologize for not clarifying in the original manuscript that we computed facial similarity using the InsightFace toolkit (https://github.com/deepinsight/insightface).
>
> The specific calculation process consists of three steps: First, we detect and crop faces in the images via InsightFace. Next, the cropped facial regions are fed into a pre-trained ArcFace model to extract 512-dimensional facial feature vectors, which are then normalized by L2 regularization. Finally, the cosine similarity between the feature vectors of the reference image and the generated image is calculated as the quantitative result of facial similarity.
>
> In essence, the facial similarity metric we adopted is fundamentally consistent with the ArcFace-based evaluation method you suggested, as it leverages the pre-trained ArcFace model within the InsightFace toolkit for discriminative feature extraction and similarity calculation. We will supplement this technical detail in the revised manuscript to enhance the clarity.

---

> ### Author Response · Authors · 2025-11-20
> **Reponse to Reviewer 3epi (Part2)**
>
> **W3: Explanation on PID algorithm**
>
> Thank you for your insightful question regarding the correspondence between the mathematical formulation of the PID control algorithm and the target values. We highly appreciate your rigorous consideration and clarify the definition of the error term $e_k$ and its alignment with the physical meaning of errors for each task as follows:
>
>   * For the ID-preserving portrait generation task, $e_k$ is defined as the element-wise difference between the reference facial feature vector and the generated facial feature vector in the k-th iteration. Both vectors are 512-dimensional discriminative feature embeddings extracted via the ArcFace model integrated in the InsightFace toolkit. The element-wise difference directly reflects discrepancies across each dimension of the identity feature space—each dimension encodes specific facial identity attributes—thus enabling $e_k$ to quantitatively capture the physical difference in identity consistency between the generated result and the reference image.
>
>   * For the pose-controlled generation task, $e_k$ corresponds to the coordinate difference of corresponding human keypoints (detected via OpenPose) between the reference image and the generated image in the k-th iteration. Specifically, we calculate the pairwise (x, y) coordinate differences of key body joints (e.g., shoulders, elbows, knees) in the two images. This coordinate difference directly represents the pixel-level positional offset of the pose, a standard physical metric for quantifying pose deviation in computer vision. The magnitude of $e_k$ accurately reflects the degree of inconsistency between the generated pose and the reference pose.
>
>   * For the depth-controlled generation task, $e_k$ is defined as the grayscale difference at corresponding positions between the reference depth map and the generated depth map in the k-th iteration. The depth maps are extracted using the MiDaS estimator, where grayscale values have a linear mapping relationship with actual depth distances. The grayscale difference at each pixel corresponds to the actual depth deviation at that position, and $e_k$ collectively quantifies the spatial depth inconsistency between the two images.
>
> In essence, the definition of $e_k$ for each task strictly aligns with the physical nature of the target error: it directly originates from differences in task-specific core metrics (identity feature dimensions, keypoint coordinates, depth grayscale values) between the reference and generated results. Each component of $e_k$ has a clear physical interpretation, ensuring its value accurately quantifies the actual deviation from the target. This design guarantees that the PID controller adjusts the generation process based on meaningful error signals, thereby effectively guiding the iterative optimization toward convergence to the desired target.
>
> **W4: Explanation on the fairness of the comparison**
>
> Thank you for your valuable comment regarding the fairness of the comparison between our method and the baseline models. We highly appreciate your consideration of the multi-round computation characteristic of our algorithm and clarify the rationality and fairness of this comparative evaluation as follows:
>
> The core objective of our proposed CL-GEN method is to enhance reference consistency through a closed-loop iterative optimization framework. This multi-round computation design is not an irrelevant "additional advantage" but the core technical innovation of our work—our original intention is precisely to address the limitation of existing methods, which can only generate results via a single forward pass and lack the capability for iterative refinement.
>
> It is important to emphasize that all previous methods do not possess multi-round iterative optimization capabilities: their generation process relies on one-time forward inference based on input references and prompts, with no mechanism to adjust results according to the deviation between outputs and references. Therefore, the comparison between our method and other approaches is essentially a comparison between two distinct technical paradigms—"single-pass generation" and "closed-loop iterative optimization generation"—both targeting the same goal of improving reference consistency. By comparing the final generation performance under the same input conditions and evaluation criteria, we can effectively demonstrate the superiority of the closed-loop iterative optimization mechanism in enhancing reference consistency. This comparative design is both fair and meaningful, as it directly reflects the value of our proposed technical innovation.
>
> **W5: Supplementation of Relevant References**
>
> Thanks for your advice. We have supplemented relevant references related to the PID control algorithm.
>
> Thank you again for devoting your time and effort to refining this paper—your insights have made it more robust and thorough.
>
> Best wishes,
>
> Team of 2798

---

> > ### Comment · Reviewer_3epi · 2025-11-26
> >
> > Thanks for the rebuttal. The rebuttal addressed part of my concerns.

---

> > > ### Author Response · Authors · 2025-11-26
> > > **Thanks Reviewer 3epi**
> > >
> > > Dear Reviewer 3epi,
> > >
> > > Thank you very much for your feedback on our rebuttal and for taking the time to review our responses carefully.
> > >
> > > If you have any further questions or additional concerns that need to be addressed, please do not hesitate to let us know. We are more than willing to provide further clarifications or make additional revisions to our work.
> > >
> > > Best wishes,
> > >
> > > Team of 2798

---

### Official Review · Reviewer_ezgE · 2025-10-30

**Soundness:** 1
**Presentation:** 2
**Contribution:** 2
**Rating:** 2
**Confidence:** 4

**Summary:**

This paper proposes CL-GEN, a reference-guided image generation framework based on inference-time optimization. The core contribution of this work is modeling reference consistent image generation by drawing an analogy between control systems and generative processes. Based on a PID-like optimization target, CL-GEN can achieve image generation results more faithful to the reference image. Experimental results on several datasets and various tasks demonstrate the effectiveness of CL-GEN.

**Strengths:**

Although the review criteria call for comments on originality, quality, clarity, and significance, I see substantive strengths only in originality; therefore, this section focuses solely on that aspect. To the best of my knowledge, framing reference-consistent image generation through the lens of control theory is novel and could motivate new inference-time optimization algorithms that offer improved output quality and finer controllability; it may also inspire researchers in related fields.

**Weaknesses:**

1. The motivation is unclear. At the beginning of the Introduction section, the authors review the task of image generation and ID-preserving image generation, and then point out the central issue of existing studies: failing to guarantee reference consistency, as well as lacking theoretical foundations. Then, the authors immediately introduce the analogy between control systems and generative processes. How does it relate to the core issue just identified? What motivates you to propose such modeling to solve the problem pointed out beforehand?

2. How does the proposed PID-like optimization framework work with the diffusion-based generation model? The tutorial in the preliminary section is insufficient, and the introduction in the method is also unclear.

3. From my perspective, the experimental results fail to demonstrate the universal advantage of CL-GEN. Based on the qualitative results shown in Figure 2, 3, and 5, honestly, I can hardly identify the advantage brought by the proposed method. This is also reflected by the quantitative values shown in these figures.

4. From Table 1, it can be clearly seen that incorporating CL-GEN can only bring subtle improvement (1e-2 ~1e-3), and sometimes even worse results. How do you explain it?

5. Since inference-time optimization will inevitably introduce extra computation cost, an in-depth analysis on the computational complexity is necessary.

**Questions:**

From my perspective, there is much room for further improvement before this manuscript can reach the threshold of being accepted by ICLR. Please refer to the 'Weakness' section for potential further improvement directions, and I do not think my evaluation and rating of this study will further change.

---

> ### Author Response · Authors · 2025-11-20
> **Reponse to Reviewer ezgE (Part1)**
>
> We would like to thank you for taking the time to review our work. Your questions and comments are precious. We value each of them and have provided detailed responses below：
>
> **W1: Revise Introduction: Clarify Research Motivation**
>
> Thank you for pointing out our issues. We have revised the Introduction section to articulate the research motivation more clearly. The following presents a comprehensive explanation of the research motivation for our work.
>
> In closed-loop control systems such as motor speed control, users specify a target speed as the reference. Sensors monitor the motor’s actual speed in real time, and the controller adjusts the control signal based on the speed deviation to drive the actual speed toward convergence with the target speed. In contrast, open-loop systems only respond to the reference without such correction capabilities. This paradigm inspires us. For controllable image generation tasks that require high reference consistency, including identity-preserving portrait generation, it is necessary to maximize the alignment between the features of generated images and the conditional input. This is analogous to classical control problems like motor control, where the conditional input acts as the reference similar to the target speed, and the controllable image generation model serves as the control plant comparable to the motor. We wonder whether a closed-loop system can be constructed by adding sensing and control modules to the existing framework. The system would leverage feedback on the feature representation of generated images for correction, driving it to converge toward the conditional input and thereby enhancing reference consistency.
>
> To validate the above assumption, we built closed-loop systems tailored to three representative tasks: identity-preserving portrait generation, pose-controlled generation, and depth-controlled generation. The proposed systems yield improved reference consistency, confirming the feasibility of our assumption. Through this work, we hope to motivate more cross-domain knowledge transfer from control theory to image generation.
>
> **W2: Supplement the Preliminary section and clarify the Method section**
>
> Following your suggestion, we have added an introduction to the basic concept of closed-loop control in the Preliminary section (Sec. 3.1), refined the description in the Method section (Sec. 4.1) regarding how closed-loop control integrates with the generative model.
>
> In Sec. 3.1, we supplement the introduction to the composition of the closed-loop control system and the functions of its key components.
>
> In Sec. 4.1, we elaborate on the specific workflow of the iterative process. In the proposed framework, the conditional input serves as reference, which defines the target value that the control system aims to achieve for the controlled plant’s output. In one iteration, the sensor first observes the previously generated image and extracts its feature representation. The controller calculates a new input based on the deviation between the reference and the actual features. The generative model, acting as the control plant, responds to this input and generates a new image. Through multiple iterations, the features of the generated images are brought closer to the reference.

---

> ### Author Response · Authors · 2025-11-20
> **Reponse to Reviewer ezgE (Part2)**
>
> **W3 & W4: Explanation on the universal advantage**
>
> Thank you for your valuable comments on the experimental results. We highly appreciate your candid feedback and provide supplementary explanations regarding the general advantages of CL-GEN as follows:
>
>   * regarding the significance of the quantitative improvements: In the field of portrait generation, performance gains on the order of 1e-2 are widely recognized as significant and practically meaningful. Numerous works published in top conferences have demonstrated the advancement of their methods through improvements of a similar magnitude. For instance, PuLID [1], a representative method in identity-preserving portrait generation, increased the facial similarity metric by 0.08 compared to its baseline InstantID [2]—rising from 0.725 to 0.733—which aligns with the improvement scale of our method. It is worth noting that the baseline model we adopted already exhibits strong performance, making subsequent incremental improvements significantly more challenging.
>
>   * our primary goal is to enhance reference consistency (i.e., facial similarity) without compromising other key aspects such as image quality and facial diversity. Facial pose diversity is a critical yet often overlooked core requirement in portrait generation tasks. While improving facial similarity, we pay special attention to avoiding rigid "copy-paste" artifacts of faces in generated images and ensuring the natural diversity of synthesized portraits. To verify this advantage, we introduced the face diversity metric for comprehensive evaluation. As shown in Table 1, our method not only achieves improved facial similarity but also maintains comparable diversity performance.
>
>   * We understand that due to the limitations of the paper’s presentation format, some optimizations are reflected in facial details, resulting in qualitative differences that may not be sufficiently prominent in intuitive visual perception. We will provide images before and after optimization in the supplementary materials. By switching between these images in an image viewer, you can more intuitively perceive the changes.
>
> [1] Zinan Guo, Yanze Wu, Zhuowei Chen, Lang Chen, Peng Zhang, and Qian He. Pulid: Pure and lightning id customization via contrastive alignment. In Advances in Neural Information Processing Systems, 2024.
>
> [2] Qixun Wang, Xu Bai, Haofan Wang, Zekui Qin, Anthony Chen, Huaxia Li, Xu Tang, and Yao Hu.
> Instantid: Zero-shot identity-preserving generation in seconds. arXiv preprint arXiv:2401.07519,
> 2024.
>
> **W5: Addition of computational efficiency analysis**
>
> Following your suggestions, we have added an analysis of computational efficiency in Section 5.5. Specifically, we conducted a computational efficiency analysis for identity-preserving portrait generation and pose-controlled generation (pose control and depth control are similar in terms of efficiency). The computing resource usage during the closed-loop optimization process of the two tasks is shown in the Figure 7. For identity-preserving portrait generation, the average iteration time per run is 4.2 seconds, with a peak video memory usage of 15,286 MB and a peak CPU utilization rate of 76.6%. For pose-controlled generation, the average iteration time per run is 11.6 seconds, accompanied by a peak video memory usage of 14,148 MB and a peak CPU utilization rate of 49.7%. The total generation time depends on the setting of the maximum number of iterations, and the time and computing resource consumption of each iteration are basically consistent with those of a single round generation.
>
> Once again, we sincerely express our best gratitude for your effort and time.
>
> Best wishes,
>
> Team of 2798

---

### Official Review · Reviewer_XvRU · 2025-10-31

**Soundness:** 3
**Presentation:** 2
**Contribution:** 2
**Rating:** 4
**Confidence:** 4

**Summary:**

This paper introduces a novel control-theoretic perspective on controllable image generation by formulating it as a closed-loop feedback system. Through a P(ose) I(d) D(epth)-based iterative optimization during inference, CL-GEN improves reference consistency without retraining. The framework is simple, generalizable, and empirically effective across ID, pose, and depth control tasks.

**Strengths:**

Strengths:
- First attempt to apply closed-loop control (PID feedback) to image generation at inference time
- Integrates control theory with diffusion-based generative modeling, offering a new theoretical lens
- No need to additional training

**Weaknesses:**

Weaknesses:
- No analysis of stability, or control gain (K_p, K_i, K_d) sensitivity is provided
- Insufficient performance comparison with SOTA methods qualitatively and quantitatively
- No computational cost analysis

**Questions:**

- Why didn't you compare it with various SOTA methods?
- In Table 1, I think that there are no significant differences across them except for facial similarity. But, the area occupied by the face in the image is not that large.
- Are there any results from generating other objects (not human) or landscapes? Will it still work like ControlNet?

---

> ### Author Response · Authors · 2025-11-20
> **Reponse to Reviewer XvRU (Part1)**
>
> We sincerely appreciate your review. Your effort has ensured that our submission received adequate attention and thorough review, and also provided us with lots of suggestions . We value each of your suggestions and provide the following responses:
>
> **W1: Addition of coefficient stability analysis**
>
> Thank you for the suggestion. We have added a coefficient stability analysis in Section C of the Appendix. We use IPA as the baseline and randomly select 25 groups of data from the Web100 dataset to test the stability of the three coefficients($K_p$, $K_i$ , $K_d$) in the closed-loop optimization method. Specifically, one parameter is adjusted while the other two are fixed, and the changes in the average facial similarity are observed. The base parameters are set as $K_p = 0.3$, $K_i = 0.05$, and $K_d = 0.01$. The results indicate that the three coefficients are effective within a wide range.
>
> **W2 & Q1: Addition of SOTA comparison**
>
> Thank you for your valuable suggestions.
>
> The primary goal of this paper is not to outperform state-of-the-art (SOTA) methods on specific tasks, but to explore whether controllable image generation can be formulated as a control problem and optimized by applying control theory. Taking three tasks—ID-preserving portrait generation, pose-controlled generation, and deep-controlled generation—as examples, we verify our hypothesis by constructing controllable image generation as a typical closed-loop control system. This system forces the features of generated images to converge toward conditional inputs, thereby achieving improved reference consistency.
>
> Your suggestions are highly valuable. Comparing with the latest SOTA methods will help demonstrate the advantages of our approach. We have added a comparison with PuLID [1]. Quantitative evaluation results are included in Table 1, and qualitative comparisons of mainstream methods (Figure A-6), which cover PuLID, have been added in the Appendix.
>
> [1] Zinan Guo, Yanze Wu, Zhuowei Chen, Lang Chen, Peng Zhang, and Qian He. Pulid: Pure and lightning id customization via contrastive alignment. In Advances in Neural Information Processing Systems, 2024.
>
> **W3: Addition of computational efficiency analysis**
>
> Following your suggestions, we have added an analysis of computational efficiency in Section 5.5. Specifically, we conducted a computational efficiency analysis for identity-preserving portrait generation and pose-controlled generation (pose control and depth control are similar in terms of efficiency). The computing resource usage during the closed-loop optimization process of the two tasks is shown in the Figure 7. For identity-preserving portrait generation, the average iteration time per run is 4.2 seconds, with a peak video memory usage of 15,286 MB and a peak CPU utilization rate of 76.6%. For pose-controlled generation, the average iteration time per run is 11.6 seconds, accompanied by a peak video memory usage of 14,148 MB and a peak CPU utilization rate of 49.7%. The total generation time depends on the setting of the maximum number of iterations, and the time and computing resource consumption of each iteration are basically consistent with those of a single round generation.

---

> ### Author Response · Authors · 2025-11-20
> **Reponse to Reviewer XvRU (Part2)**
>
> **Q2: Explanation on quantitative evaluation on ID-preserving portrait generation.**
>
> The primary goal of this paper in the identity-preserving portrait generation task is to maximize the similarity between the face in the generated image and that in the reference image. Facial similarity is computed using facial feature vectors optimized via ArcFace’s [2] angular margin loss, which ensures high inter-class discriminability. The resulting similarity can directly quantify the identity consistency between the generated and reference faces, perfectly aligning with the core demand of the task. This metric has become a universal evaluation standard in both academia and industry.
>
> Building on this, we adopt CLIP-I and DINO metrics to measure the semantic similarity between the generated and reference faces, and Q-Align to assess the quality of generated images. Notably, the pose of the generated portrait should be flexibly controllable to avoid the "copy-paste" phenomenon. While "copy-paste" significantly boosts facial similarity, it deviates from the original intent of the task. Therefore, we propose the structure diversity metric to evaluate the flexibility and diversity of the generated portrait poses.
> The results in Table 1 demonstrate that the closed-loop optimization method effectively improves facial similarity while maintaining other metrics at a level comparable to baseline models. For instance, our method does not sacrifice structure diversity for enhanced facial similarity.
>
> [2]Deng, Jiankang, et al. Arcface: Additive angular margin loss for deep face recognition. Proceedings of the IEEE/CVF conference on computer vision and pattern recognition. 2019.
>
> **Q3: Addition of non-human results**
>
> ID-preserving portrait generation and pose-controlled generation are human-related tasks, thus non-human samples cannot be tested on them. In contrast, deep-controlled generation is irrelevant to the presence of humans in reference images, so we have added tests on non-human samples for this task. The relevant results are presented in Figure A-5 of the Appendix.
>
> Thank you again for devoting your time and effort to refining this paper—your insights have made it more robust and thorough.
>
> Best wishes,
>
> Team of 2798

---

### Author Response · Authors · 2025-11-20
**Response Summary and Paper Revision**

Dear Reviewers,

We sincerely thank you for the valuable feedback, which has significantly improved the quality and rigor of our paper. Based on your suggestions, we have revised and updated the paper, highlighting changes and additions in red within the revised PDF. Below, we summarize the key revisions made:

1. Clarify Research Motivation
Based on the comments from Reviewer ezgE, we have revised the Introduction section to clarify the research motivation and core focus of this paper. Our motivation can be summarized as follow.
Many applications require strict consistency between generated results and reference conditions. Inspired by control theory—where closed-loop systems use continuous feedback to drive the stable convergence of outputs to target values—this paper aims to formulate controllable image generation as a control problem. It achieves this by constructing a closed-loop system to force the features of generated images to converge toward conditional inputs, thereby enhancing reference consistency.
2. Supplement the Preliminary section and clarify the Method section
Following the suggestions from Reviewers ezgE, 3epi, and QLbv, we have added an introduction to the basic concept of closed-loop control in the Preliminary section, refined the description in the Method section regarding how closed-loop control integrates with the generative model, and further explained the rationale for revising Equation 1 to Equation 3.
3. Addition of SOTA comparison
Following the suggestions from Reviewers XvRU and 3epi, we have added a comparison with PuLID. Quantitative evaluation results are included in Table 1, and qualitative comparisons of mainstream methods (Figure A-6), which cover PuLID, have been added in the Appendix.
4. Addition of computational efficiency analysis
Following the suggestions from Reviewers ezgE and QLbv, we have added an analysis of computational efficiency in Section 5.5. Compared with the baseline methods, the peak memory usage and CPU utilization do not increase after adding closed-loop optimization. After loading the model and data, the total inference time is approximately the single inference time multiplied by the number of iterations.
5. Addition of coefficient stability analysis
Following the suggestion from Reviewer XvRU, we have added a coefficient stability analysis in Section C of the Appendix. The results indicate that the three coefficients, $K_p$, $K_i$, and $K_d$, are effective within a wide range.
6. Addition of non-human results
Following the suggestion from Reviewer XvRU, we have conducted tests on non-human samples in the deep controlled generation task, with the visualization results presented in Figure A-5 of the Appendix.
7. Supplementary materials
In identity-preserving portrait generation, due to the limitations of the paper’s presentation format, some optimizations are reflected in facial details, making qualitative differences less prominent in intuitive visual perception. We provide images before and after optimization in the supplementary materials; switching between these images in an image viewer allows for a more intuitive perception of the changes. Additionally, to address Reviewer QLbv’s concern that our method can be implemented with only a few lines of code, we have included the source code before and after modifications in the supplementary materials.

Once again, we deeply appreciate the time and effort you have dedicated to improving this paper. Your insights have made the paper stronger and more comprehensive.

Best wishes,

Team of 2798

---

### Meta-Review · Area_Chair_XR4A · 2025-12-13

**Summary:**

The paper proposes CL-GEN, a controllable image generation framework that formulates reference-guided generation as a closed-loop control problem inspired by classical control theory. Instead of relying on a single forward pass, CL-GEN introduces an inference-time iterative optimization loop using a PID-like controller to reduce the discrepancy between generated images and reference conditions.


Here are the summary of reviewer concerns:
1.Conceptual & Theoretical Clarity
Motivation initially unclear: how control theory directly addresses reference consistency
Concern that the control-theoretic framing might be over-packaged for what appears to be iterative feedback

2. Experimental Results are not strong and miss SOTA comparisons like Added non-human results to test generalization

**Reviewer Concerns:**

Here are some high-level summarize:

What has been addressed:
Most of the do-able concerns like adding more explanations to motivations, adding more   Experimental Results/running efficients analysis has been added to the rebuttal.

What are still outstanding:
he main outstanding issues are not missing experiments or unclear methodology, but rather whether the empirical gains, generality, and practical impact are strong enough to justify the closed-loop control framing and iterative inference overhead for a broad ICLR audience.

**Reviewer Scores:**

two reviewers are 4 – Marginally below acceptance
two reviewers are 2 – Reject

---

### Decision · Program_Chairs · 2026-01-26

Reject